# Strontium Carbonate and Strontium-Substituted Calcium Carbonate Nanoparticles Form Protective Deposits on Dentin Surface and Enhance Human Dental Pulp Stem Cells Mineralization

**DOI:** 10.3390/jfb13040250

**Published:** 2022-11-17

**Authors:** Tatiane Cristina Dotta, Larwsk Hayann, Leonardo de Padua Andrade Almeida, Lucas Fabrício B. Nogueira, Mayara M. Arnez, Raisa Castelo, Ana Flávia B. Cassiano, Gisele Faria, Milena Martelli-Tosi, Massimo Bottini, Pietro Ciancaglini, Alma B. C. E. B. Catirse, Ana Paula Ramos

**Affiliations:** 1Department of Dental Materials and Prosthodontics, Ribeirão Preto School of Dentistry, University of São Paulo, Ribeirão Preto 14040-904, Brazil; 2Department of Chemistry, Ribeirão Preto Faculty of Philosophy, Sciences and Letters at Ribeirão Preto, University of São Paulo, Ribeirão Preto 14040-901, Brazil; 3Department of Restorative Dentistry, School of Dentistry at Araraquara, Sao Paulo State University (UNESP), Araraquara 14801-385, Brazil; 4Department of Food Engineering, Faculty of Animal Science and Food Engineering, University of São Paulo, Pirassununga 13645-900, Brazil; 5Department of Experimental Medicine, University of Rome Tor Vergata, 00133 Rome, Italy

**Keywords:** biomaterials, dental hypersensivity, dentin, mineralization, strontium-containing nanoparticles

## Abstract

Strontium acetate is applied for dental hypersensitivity treatment; however, the use of strontium carbonates for this purpose has not been described. The use of Sr-carbonate nanoparticles takes advantage of both the benefits of strontium on dentin mineralization and the abrasive properties of carbonates. Here in, we aimed to synthesize strontium carbonate and strontium-substituted calcium carbonate nanoparticles and test them as potential compounds in active dentifrices for treating dental hypersensitivity. For this, SrCO_3_, Sr_0.5_Ca_0.5_CO_3_, and CaCO_3_ nanoparticles were precipitated using Na_2_CO_3_, SrCl_2_, and/or CaCl_2_ as precursors. Their morphology and crystallinity were evaluated by electron microscopy (SEM) and X-ray diffraction, respectively. The nanoparticles were added to a poly (vinyl alcohol) gel and used to brush dentin surfaces isolated from human third molars. Dentin chemical composition before and after brushing was investigated by infrared spectroscopy (FTIR) and X-ray dispersive energy spectroscopy. Dentin tubule morphology, obliteration, and resistance of the coatings to acid attack were investigated by SEM and EDS. The cytotoxicity and ability of the particles to trigger the mineralization of hDPSCs in vitro were studied. Dentin brushed with the nanoparticles was coated by a mineral layer that was also able to penetrate the tubules, while CaCO_3_ remained as individual particles on the surface. FTIR bands related to carbonate groups were intensified after brushing with either SrCO_3_ or Sr_0.5_Ca_0.5_CO_3_. The shift of the phosphate-related FTIR band to a lower wavenumber indicated that strontium replaced calcium on the dentin structure after treatment. The coating promoted by SrCO_3_ or Sr_0.5_Ca_0.5_CO_3_ resisted the acid attack, while calcium and phosphorus were removed from the top of the dentin surface. The nanoparticles were not toxic to hDPSCs and elicited mineralization of the cells, as revealed by increased mineral nodule formation and enhanced expression of COL1, ALP, and RUNX2. Adding Sr_0.5_Ca_0.5_CO_3_ as an active ingredient in dentifrices formulations may be commercially advantageous since this compound combines the well-known abrasive properties of calcium carbonate with the mineralization ability of strontium, while the final cost remains between the cost of CaCO_3_ and SrCO_3_. The novel Sr_0.5_Ca_0.5_CO_3_ nanoparticles might emerge as an alternative for the treatment of dental hypersensitivity.

## 1. Introduction

Loss of dental structure related to diet, incorrect tooth brushing, aging, and occlusal disorders, among other factors, can cause dental hypersensitivity. This disorder, which is characterized by intense pain after osmotic, thermal, or chemical stimulus, affects up to 30% of the global population [1,2,3,4,5,6,7,8]. The hydrodynamic theory proposed by Brännström in the 1960s explains that hypersensitivity is associated with dentin fluid reaching nerve fibers located either inside or at the interface of dentin tubules, thereby triggering pain [3,4,5,6,9,10,11]. Several treatments have been proposed for hypersensitivity. Among them, the use of dentifrices containing strontium acetate, arginine, potassium nitrate, or tin monofluorophosphate fluoride as a desensitizing agent is the most common because they are relatively inexpensive and can be applied at home [6,8,10,12,13,14,15]. Desensitizing agents can also be used to prevent tooth erosion caused by ingestion of foods and drinks with acid pH, including citrus fruits, wine, and fruit juice, or even by stomach pH deregulation [16,17,18,19,20,21]. 

Strontium, especially from water and food, is naturally incorporated into mineralized tissues such as bone and teeth because Sr^2+^ can substitute Ca^2+^ in the apatite structure [22]. In this sense, Sr^2+^ is also associated with calcified tissue remineralization [2,22,23,24,25,26]. Strontium chloride was the first strontium-based desensitizing agent to be applied in hypersensitivity treatment [27,28]. However, there was no clinical evidence of the positive effects of this compound as compared to the fluorides employed for the same purpose. Two theories explain the positive effects of strontium during hypersensitivity treatment: (a) incorporation of Sr^2+^ into vacant Ca^2+^ sites due to demineralization and reprecipitation of strontium-apatite mineral [2,14,29,30,31]; and (b) precipitation of organic compounds of the dental matrix and odontoblastic activation, to form a blocking layer that prevents liquids from circulating inside dentin tubules [31]. Nowadays, dentifrices contain strontium as the acetate salt [27,32], but other strontium-containing compounds have also been applied for bone tissue remineralization. Strontium ranelate is one example, but its use has been suspended almost worldwide due to some side effects, particularly those related to cardiovascular diseases [24]. In turn, strontium carbonate positively affects bone regeneration, so it has been associated with a reduced risk of fracture in healthy and osteoporotic animal models and with accelerated osteointegration of dental implants [33]. Despite these advantages, strontium carbonate has not yet been included in the formulation of dentifrices yet [33,34,35,36]. Using strontium carbonate and strontium/calcium carbonate in dentifrices would allow the well-known abrasive effect of calcium carbonate and the remineralizing and desensitizing properties of strontium to be combined. Moreover, compared to strontium acetate, the lower solubility of strontium carbonate would prevent Sr^2+^ removal from the dentin surface. For instance, Seong et al. have found augmented pain alleviation in patients treated with potassium nitrate-mixed hydroxyapatite compared to the pure potassium nitrate toothpaste, which is also a highly soluble salt [37].

More recently, the action of Sr^2+^ on dental mineralization at a cell level has also been studied [38,39,40]. Martin-del-Campo et al. prepared 3D scaffolds containing strontium folate embedded with human dental pulp stem cells (hDPSCs) as mineralizing cell models [39]. Cell differentiation to a mineralizing phenotype was stimulated by the presence of Sr^2+^. The mechanisms of action of Sr^2+^ released from bioactive glasses on the proliferation and differentiation of hDPSCs have been explained through the stimulus of the expression of key odontogenic markers, including dentine sialophosphoprotein (DSPP) and dentine matrix protein 1 (DMP-1) [38,40].

Here, we aimed to synthesize strontium carbonate and strontium-substituted calcium carbonate nanoparticles and testing them as potential compounds in active dentifrices for treating dental hypersensitivity. The use of nanoparticles allowed strontium deposition on the dentin surface and penetration into dentin tubules, as well as tubule obliteration. Moreover, the particles induced osteogenic differentiation of hDPSCs. Strontium carbonate may promote dentin surface abrasion and remineralization simultaneously. The new mineral coatings deposited at the dentin surfaces after the treatment were resistant to acid attack.

## 2. Materials and Methods

### 2.1. Materials

Dentin discs were prepared from third molar teeth from the tooth bank of the Dental School (University of São Paulo, Ribeirão Preto, Brazil) after approval of the Ethics in Research Committee of the same institute (06245218.2.0000.5419). To synthesize SrCO_3_ and Sr_0.5_Ca_0.5_CO_3_ nanoparticles, calcium chloride, strontium chloride, and sodium carbonate were acquired from Labsynth and used without further purification. Poly (vinyl) alcohol (MW 9000–10,000, 80% hydrolyzed) was purchased from Sigma-Aldrich. All the aqueous solutions were prepared with ultrapure deionized water from a Milli-Q filtration system (resistivity 18.2 M·Ω·cm). 

### 2.2. Methods

#### 2.2.1. Preparation of Dentin Discs

To obtain 1 mm thick dentin discs, the third molars were sectioned at the cementoenamel junction; an acrylic template (Dencor^®^, Methyl Methacrylate, *Artigos Odontológicos Clássico* Ltda, São Paulo, Brazil) was used as sample holder. The best dentin discs were selected with the aid of a magnifying glass; discs containing tertiary dentin, caries, and cracks were eliminated [41]. The dentin discs were cleaned in deionized water in an ultrasound bath (ALTSonic Clean- ALT Equipamentos, Ribeirão Preto, Brazil) for 30 s. Then, the discs were kept in 6 wt.% aqueous citric acid solution for 2 min to remove the dental slurry [41]. Next, the discs were immersed in deionized water and cleaned in an ultrasound bath for 10 min. After cleaning, all the discs were conditioned in 0.5 mL of artificial saliva [42] and maintained at 37 ± 1 °C.

#### 2.2.2. Synthesis of SrCO_3_, Sr_0.5_Ca_0.5_CO_3_, and CaCO_3_ Nanoparticles

SrCO_3_ nanoparticles were prepared by mixing 100 mL of 2.66 mol·L^−1^ aqueous SrCl_2_ solution and 100 mL of 1.06 mol·L^−1^ Na_2_CO_3_. The formation of a white precipitate evidenced that a poorly soluble product assigned to SrCO_3_ was produced. The reaction is depicted below:SrCl_2_(aq) + Na_2_CO_3_ (aq) → SrCO_3_(s) + 2NaCl(aq)(1)

Sr_0.5_Ca_0.5_CO_3_ nanoparticles were obtained by mixing 100 mL of 1.33 mol·L^−1^ aqueous SrCl_2_ solution and 100 mL of 0.73 mol·L^−1^ CaCl_2_. Then, 100 mL of 1.06 mol·L^−1^ aqueous Na_2_CO_3_ solution was added to this mixture. The formation of a white precipitate evidenced that a poorly soluble product assigned to SrCO_3_ was produced. The reaction is depicted below:½ SrCl_2_(aq) + ½ CaCl_2_(aq) + Na_2_CO_3_ (aq) → Sr_0.5_Ca_0.5_CO_3_(s) + 2NaCl(aq)(2)

CaCO_3_ nanoparticles were prepared by mixing 100 mL of 1.47 mol·L^−1^ aqueous CaCl_2_ solution and 100 mL of 1.06 mol·L^−1^ Na_2_CO_3_.
CaCl_2_(aq) + Na_2_CO_3_ (aq) → CaCO_3_(s) + 2NaCl(aq)(3)

All the reactions were carried out under vigorous stirring at 20,000 rpm for 10 min on an Ultra Turrax^®^ device (IKA^®^ T18 basic, IKA^®^ Works Brazil Ltda, Campinas, Brazil). The solids were filtered, washed three times with deionized water for purification, and dried under controlled conditions at 50 °C for 40 h before being characterized and applied in further experiments. 

#### 2.2.3. Characterization of Nanoparticles

The crystalline structure of the SrCO_3_, Sr_0.5_Ca_0.5_CO_3_, and CaCO_3_ nanoparticles was investigated by X-ray diffraction (XRD) in a D2 PHASER, Bruker-AXS powder diffractometer, with CuKα as source (30 kV, 10 mA). The *Crystallography Open Database* (COD) was used to index the peaks. The size and the charge of the nanoparticles dispersed in an aqueous medium were investigated by dynamic light scattering (DLS) in a ZetaSizer Nano device (Malvern, UK) equipped with a 580 nm laser at a scattering angle of 172°. The morphology of the gold-coated nanoparticles was studied by scanning electron microscopy (SEM) on a microscope (Shimadzu SS-550, Kyoto, Japan) operating at 20 kV.

#### 2.2.4. Preparation of Gels Containing Nanoparticles for Application on Dentin Discs

First, an aqueous gel containing 10 wt.% PVA was prepared by dissolving the polymer in ultrapure deionized water under mechanical stirring at 90 °C. Then, 30 wt.% SrCO_3_, Sr_0.5_Ca_0.5_CO_3_, or CaCO_3_ nanoparticles were added to this gel under stirring at 25 °C until a homogeneous gel was obtained. The rheological behavior was analyzed with a Modular Compact Rheometer MCR 52 (Antom-Paar GmbH, Graz, Austria) using the cone and plate (CP50-1, 50 mm, angle 1°) geometry. Steady-state flow measurements were carried out with shear rates ranging from 1 to 100 s^−1^ at 25 °C ± 3. The samples presented pseudoplastic behavior, and the apparent viscosity ranged from 2.2 up to 2.9 Pa.s (at 100 s^−1^) without statistical differences. 

#### 2.2.5. Application of Gel Containing Nanoparticles to Dentin Surface by Brushing

A gel containing SrCO_3_, Sr_0.5_Ca_0.5_CO_3_, or CaCO_3_ nanoparticles (0.3 g) was deposited on the surface of the dentin discs. Then, the discs were coupled to a mechanical brushing simulator (Biopdi, São Carlos, Brazil) and positioned perpendicularly to the toothbrush bristle under a force corresponding to 200 g. Each brushing cycle involved 40 movements at the machine at a rate of 4.5 movements/s^−1^, in a total of 10 cycles/s^−1^. After brushing, the discs were washed with deionized water for 10 s and immersed in artificial saliva at 37 °C for 4.5 h until the next brushing [16]. This procedure was repeated twice a day for 14 days. The discs were divided Into four groups, according to the nanoparticle that was tested as a desensitizing agent, namely SrCO_3_, Sr_0.5_Ca_0.5_CO_3_, CaCO_3_, and Sr(CH_3_COO)_2_ (Sensodyne^®^ Rapid Relief).

#### 2.2.6. Resistance of Coating to Acid Attack

The resistance of the coatings deposited on the surface of the dentin discs was tested under an acid attack promoted by immersing the discs in 3 mL of Coca-Cola^®^ for 1 h under slow stirring (50%) in a stirring table (Cintec, Ribeirão Preto, Brazil); the procedure described by Olley et al. was followed [16]. After this procedure, the discs were washed with deionized water for 10 s and stored in artificial saliva at 37 °C. 

#### 2.2.7. Chemical Characterization of Dentin Discs 

Changes in the chemical composition of the dentin surfaces before and after brushing with a desensitizing agent were followed by Fourier-transform infrared spectroscopy (FTIR); an attenuated total reflectance (ATR) accessory (IRPrestige-21, Shimadzu) was used. The presence of the chemical elements Ca, Sr, and P on the surfaces was semi-quantitatively analyzed by X-ray scattering spectroscopy (EDS) (500 Digital Processing; IXRF Systems) coupled to a scanning electron microscope (Zeiss EVO 50—Zeiss, Oberkochen, Germany). The samples were coated with a thin conductive graphite layer before the EDS analyses (Bal-Tec SCD 050 Sputter Coater).

#### 2.2.8. Obliteration of Dentin Tubules 

The morphology of gold-coated (Bal-Tec SCD 050 Sputter Coater) dentin discs was analyzed by SEM. The images were also used for the calculus of the dentin tubules obliteration after brushing with a desensitizing agent [43]. To this end, two observers analyzed the images and defined each individual tubule as partially obliterated (P), completely obliterated (O), and non-obliterated (A). The sum of P + A + O should be the total number of tubules (T) observed in the image. Therefore, the number (*n*) of obliterated tubules was calculated by Equation (4):(4)$$n=\frac{\left(P\right)+\left(O\right)}{T}\times 100$$

The average value of *n* was calculated by using the value obtained by each observer. 

#### 2.2.9. Cell Culture Conditions

Human dental pulp stem cells (hDPSCs) were used in the experiments. The pulp tissue was collected from the third molars of young patients, extracted for orthodontic reasons. Cells were isolated using the enzymatic dissociation method and analyzed by flow-cytometric immunophenotyping to confirm the mesenchymal identity (data not shown). The hDPSCs were cultured in α-minimum essential medium (α-MEM, Sigma-Aldrich, St. Louis, MO, USA), supplemented with 10% fetal bovine serum (FBS), penicillin (100 IU/mL), and streptomycin (100 µg/mL), under 5% CO_2_, in a 95% humidified atmosphere at 37 °C. The α-MEM osteogenic medium—OM (with 50 μg/mL L-ascorbic acid and 10 mM β-glycerophosphate) was used to conduct all the cell-culture-related assays. Cells from the 4th passage were used in the experiments.

#### 2.2.10. Cell Viability Measurement

To test the cytotoxicity of the nanoparticles (CaCO_3_, Sr_0.5_Ca_0.5_CO_3_, and SrCO_3_), cells were trypsinized, resuspended in α-MEM, seeded on 96-well plate at the density 1.3 × 10^4^ cells per well and incubated in the air at 37 °C and 5% CO_2_. Cells were allowed to adhere on the bottom surface of the plate for 24 h, followed by the treatment with the nanoparticles (0.1, 1, and 10 μg/mL). To assess cell viability, we applied MTT assay after 24 and 48 h of culture. Cell viability was expressed as the percentage of the average of three experiments compared to the untreated control for each day of the culture.

#### 2.2.11. Tissue Non-Specific Alkaline Phosphatase (TNAP) Activity

To determine TNAP activity, firstly, we exposed the hDPSCs (2 × 10^4^ cells/well on 24-well plate) to the nanoparticles (CaCO_3_, Sr_0.5_Ca_0.5_CO_3,_ and SrCO_3_ at 1μg/mL) for 7 days and 14 days. The medium was changed every 48 h supplemented with or without (control) the nanoparticles. Then, we harvested the plasma membrane fraction in triplicate from cells after 7 and 14 days of culture, according to (71). TNAP activity was accomplished by the degradation of p-nitrophenylphosphate (pNPP), and its subproduct was analyzed by reading its absorbance at 410 nm. The activity was expressed as U mg^−1^ of total protein content so that one unit of enzyme is defined as the amount of enzyme capable of hydrolyzing 1.0 nmol of substrate per mg of protein at 37 °C.

#### 2.2.12. Mineralized Nodule Formation

Alizarin Red staining (ARS) was used to assess the calcium deposits. The hDPSCs were plated (2 × 10^4^ cells/well) in 24-well plates. The medium with and without the nanoparticles (1 μg/mL) was renewed every two days. After 14 days, the cells were fixed with 70% ethanol at 4 °C for 1 h, rinsed with deionized water, and stained with 2% ARS (pH 4.2, Sigma-Aldrich). The nodules were dissolved with 10% cetylpyridinium chloride (Sigma-Aldrich) under shaking for 15 min. Aliquots of the resuspension were collected, and the optical density was measured at 562 nm wavelength in a spectrophotometer. The results obtained from the absorbance of the control were considered as 100%.

#### 2.2.13. Messenger RNA (mRNA) Expression by Real-Time Polymerase Chain Reaction (RT-qPCR)

The effect of the nanoparticles on the expression of osteogenic biomarkers collagen type 1 alpha 1 (COL1A1), alkaline phosphatase (ALP), runt-related transcription factor 2 (RUNX2), and glyceraldehyde 3-phosphate dehydrogenase (GAPDH) was evaluated by RT-qPCR. hDPSCs were incubated with 1 µg/mL of the nanoparticles and cultivated for 7 and 14 days. For RNA extraction, the PureLink™ RNA Mini Kit (Invitrogen, Life Technologies, Grand Island, NY, USA) was used, following the manufacturer’s instructions. Complementary DNA (cDNA) was synthesized from 1 μg of total RNA using the High-Capacity cDNA Reverse Transcription Kit (Applied Biosystems, Life Technologies, Grand Island, NY, USA). Gene expression (*n* = 3) was analyzed in StepOne equipment (Applied Biosystems) using TaqMan PCR Master Mix and probes (Applied Biosystems) for the target genes. Transcripts were normalized to GAPDH, and data were shown as relative mRNA expression using the cycle threshold (2^-ΔΔCt^) method [44].

## 3. Results

### 3.1. Characterization of SrCO_3_, Sr_0.5_Ca_0.5_CO_3_, and CaCO_3_ Nanoparticles

#### 3.1.1. Investigation of the Crystalline Structure of Nanoparticles by X-Ray Diffraction

Figure 1 depicts the X-ray diffraction pattern of the SrCO_3_, Sr_0.5_Ca_0.5_CO_3_, and CaCO_3_ nanoparticles. The reaction between Ca^2+^ and CO_3_^2−^ in an aqueous solution resulted in the precipitation of pure calcium carbonate (CaCO_3_) as calcite [COD-1010962] (Figure 1—black line). In turn, the reaction between Sr^2+^ and CO_3_^2−^ resulted in the precipitation of pure strontium carbonate as strontianite [COD-5000093] (Figure 1—green line). Precipitation of a mixed calcium-strontium carbonate mineral [COD 9015600] from the mixture of Ca^2+^ and Sr^2+^ with CO_3_^2−^ was evident in the diffraction pattern depicted in Figure 1—red line. The peaks corresponding to strontianite (Figure 1—green line) shifted to higher 2θ values, the corroborating substitution of Sr^2+^ with a lighter ion, i.e., Ca^2+^, in the structure of the mixed carbonate (Figure 1—red line). Phase segregation was not evident.

#### 3.1.2. Nanoparticle Size and Charge 

The synthesis of particles presenting a diameter smaller than the diameter of the tubules is mandatory to allow the particles to penetrate the dentin structure. Moreover, either lower negative or positive charges also might favor the deposition on the negatively charged dentin surface [45]. Table 1 lists the mean diameter of the carbonate nanoparticles. CaCO_3_ presented size distribution by number centered at 590 nm. The strontium-containing carbonate nanoparticles were the smallest among the ones we studied; their size distribution was centered at 66.7 and 387.0 nm for Sr_0.5_Ca_0.5_CO_3_ and SrCO_3_, respectively. The small diameter of Sr_0.5_Ca_0.5_CO_3_ can be assigned to their slightly higher zeta-potential (ζ) (Table 1), which may have prevented aggregation. 

#### 3.1.3. Nanoparticle Morphology 

We investigated the nanoparticles’ morphology by SEM (Figure 2). SrCO_3_ (Figure 2A) and Sr_0.5_Ca_0.5_CO_3_ (Figure 2B) nanoparticles exhibited a spherical morphology, in agreement with the DLS data (Table 1). We verified the typical orthorhombic shape for the calcite-CaCO_3_ nanoparticles (Figure 2C). We assigned the presence of aggregates (red arrows) to the drying process as well as to the low ζ value.

#### 3.1.4. Effects of the Nanoparticles on the Viability of hDPSCs 

We tested the potential cytotoxic effect of the nanoparticles using cell assays in vitro. For this, we carried MTT assay at 24 and 48 h. The data depicted in Figure 3a,b showed that the nanoparticles did not significantly affect the viability of hDPSCs in concentrations up to 10 µg/mL, except for the SrCO_3._

#### 3.1.5. TNAP Activity and Mineralized Nodules Formation

We studied the effect of the nanoparticles upon mineralization by measuring the enzymatic activity of tissue non-specific alkaline phosphatase (TNAP), the main enzyme responsible for inorganic phosphate generation in mineralization-competent cells, and quantifying the formation of mineral nodules by alizarin red staining [46,47]. The results presented in the Figure 3C revealed that TNAP enzymatic activity was not significantly affected by the presence of the particles at the 7th day of culture. However, after 14 days, a slight reduction was observed for the samples treated with Sr_0.5_Ca_0.5_CO_3_ nanoparticles. Quantification of mineral nodules revealed the ability of all the nanoparticles to stimulate mineralization by hDPSCs (Figure 3D). The formation of mineralized nodules was increased up to 30% in the presence of CaCO_3_ and Sr_0,5_Ca_0,5_CO_3_ nanoparticles and 18% in the presence of SrCO_3_ nanoparticles, compared to the control in the absence of the particles. 

#### 3.1.6. Expression of RUNX2, COL1, and ALP by hDPSCs

The expression of the main osteogenic markers RUNX2, COL1, and ALP by hDPSCs was evaluated by RT-PCR analysis. The data presented in Figure 4 revealed increased expression of COL1 (Figure 4A) and ALP (Figure 4B) genes after 7 days of culturing in the presence of all the nanoparticles, compared to the control. This result corroborates the enhanced formation of mineral nodules by the cells cultivated in the presence of the nanoparticles (Figure 3D). After 14 days of culturing, the expression of COL1 and ALP was reduced, following the fate of the cells. It is worth noting that SrCO_3_ nanoparticles, which had the highest amount of Sr, sustained the increased expression of ALP and increased the expression of the transcription factor RUNX2 after 14 days of culture, following the positive effects of strontium on mineralization [48].

### 3.2. Chemical Analysis of Dentin Surface before and after Brushing with a Desensitizing Agent

Figure 5A–C illustrate the ATR-FTIR spectra of the dentin surface before (black line) and after brushing with a desensitizing agent for 1 (red lines), 7 (green lines), or 14 days (blue lines). For comparison, we also present results for dentin discs brushed with Sensodyne^®^ Rapid Relief under the same experimental conditions (Figure 5D). 

All the discs displayed bands attributed to asymmetric stretching of the PO_4_^3−^ group at 1070 cm^−1^. Furthermore, we detected the low intense band at 1250 cm^−1^, which is related to the amide-I group of collagens and is important for the characterization of the dentin surface. Other low-intensity bands in the range of 1750 and 1400 cm^−1^, ascribed to the presence of carboxylate and amine groups in the organic matrix, emerged in the spectra. 

After brushing with SrCO_3_, the bands at 1480 and 850 cm^−1^ intensified, as indicated by the black arrows in Figure 5A. This was associated with asymmetric stretching of the CO_3_^2−^ groups, thus indicating the deposition of carbonate particles on the dentin surface. The band related to the PO_4_^3−^ group did not change significantly. We observed a similar behavior after brushing with Sr_0.5_Ca_0.5_CO_3_ nanoparticles (Figure 5B). Interestingly, brushing with CaCO_3_ nanoparticles did not alter the intensity of the bands at 1480 and 850 cm^−1^ significantly (Figure 5C), but the band at 1070 cm^−1^, assigned to PO_4_^3−^ asymmetric stretching, increased, as indicated by the red arrows in Figure 5C. The ATR-FTIR results obtained after brushing with Sensodyne^®^ Rapid Relief, well-known for its desensitizing properties, revealed the effect of this product on the structure of the phosphate groups, as demonstrated by the changes in the intensity and shape of the band at 1070 cm^−1^.

### 3.3. Morphological Characterization and Quantitative Analysis of Dentin Tubule Obliteration after Brushing with a Desensitizing Agent

We analyzed the morphology of the dentin discs before (Figure 6A_1_–A_4_) and after brushing with SrCO_3_ nanoparticles (Figure 6B), Sr_0.5_Ca_0.5_CO_3_ nanoparticles (Figure 6D), CaCO_3_ nanoparticles (Figure 6F), or Sensodyne^®^ Rapid Relief (Figure 6H). 

For the discs brushed with SrCO_3_ nanoparticles (Figure 6B) or Sr_0.5_Ca_0.5_CO_3_ nanoparticles (Figure 6), the nanoparticles were deposited at the top as well as inside the dentin tubules, which produced a coating and led to 67.7% and 93.3% tubule obliteration, respectively (Table 2). The acid attack did not remove the coatings (Figure 6C,E), and obliteration was higher after this procedure. 

After brushing with CaCO_3_ nanoparticles, micrometric particles were present at the top surface. The larger size of these particles did not allow them to penetrate the dentin tubules. Moreover, the particles were removed, and a completely clean surface emerged after acid attack (Figure 6G). 

Brushing with Sensodyne^®^ Rapid Relief (Figure 6H) promoted the deposition of a coating in the intertubular spaces of the dentin surface, but obliteration was low (19.5%) (Table 2). Although coating morphology changed after acid attack, the tubules remained open (Figure 6I).

We also acquired SEM images of transversal cuts of the dentin discs before (Figure 7A_1_–A_4_) and after (Figure 7B,D,F,G) brushing. Tubule morphology clearly changed after brushing with either the particles or Sensodyne^®^ Rapid Relief. These changes were related to the deposition of the desensitizing nanoparticles inside the tubules and their adhesion to the tubule walls. The acid attack did not remove the nanoparticles from the tubules, particularly for the discs brushed with Sr_0.5_Ca_0.5_CO_3_ nanoparticles (Figure 7E).

### 3.4. Quantification and Mapping of Ca, Sr, and P in Dentin Discs

By using EDS, we quantified and mapped the chemical elements present on the dentin disc surface and inside the tubules before and after brushing with the nanoparticles. We compared the results to data obtained after brushing with Sensodyne^®^ Rapid Relief. Table 3 summarizes the (Ca + Sr)/P molar ratio calculated for the dentin discs. The control group, consisting of non-brushed dentin discs, presented a (Ca + Sr)/P molar ratio close to 1.81, in agreement with the values reported for biological apatite. The discs brushed with SrCO_3_ nanoparticles or Sr_0.5_Ca_0.5_CO_3_ nanoparticles had a (Ca + Sr)/P ratio close to the control. However, the amount of strontium was higher in the brushed than in the non-brushed teeth, as expected. After the acid attack, the relative amount of calcium reduced significantly in the samples brushed with strountium-containing nanoparticles, which reflected in an increased amount of strontium since the calculations are relative. Therefore, strontium remained on the dentin surface even after contact with acidic fluid. Discs brushed with CaCO_3_ nanoparticles presented a lower relative amount of calcium and phosphorus, and the acid challenge reduced the amount of these elements on the surface of the discs even further because of the abrasive process. 

For the samples treated with Sensodyne^®^ Rapid Relief, the strontium content increased after brushing compared to the control as an effect for strontium acetate deposition. However, the strontium content decreased after the acid attack, which was reflected in increased calcium relative amount. Nevertheless, the decreased strontium content after the acid attack evidenced weak attachment of this element to the dentin surface when strontium acetate is used as an active ingredient.

We performed EDS mapping of Ca, Sr, and P on the surface of dentin discs brushed with strontium-containing carbonate nanoparticles before (Figure 8) and after (Figure 9) acid attack to illustrate the quantitative results presented in Table 3. It is important to note that the maps obtained from the top view analysis are a sum of the contributions of the elemental content at the surface and the depth by which the electron beam can penetrate. The images evidenced strontium deposition on the surface (Figure 8—dark blue) and inside the tubules (Figure 9—dark blue) of the dentin discs brushed with SrCO_3_ nanoparticles or Sr_0.5_Ca_0.5_CO_3_ nanoparticles. With respect to brushing with Sensodyne^®^ Rapid Relief, the presence of open tubules and low obliteration was clear and corroborated the SEM analysis (Figure 6). More interestingly, observation of the transversal cut of the dentin demonstrated calcium and phosphorus removal (Figure 9—green and red) from the top surface after the acid attack for all the dentin samples. However, strontium remained on the surface (Figure 9—dark blue), revealing the specific binding of this element to the discs and the resistance of the coatings to acidic conditions.

## 4. Discussion

Different active ingredients and several methodologies have been proposed to treat dental hypersensitivity [5,7,49]. Most treatments involve blocking the exposed dentin tubules and desensitizing the pulp nerve. However, the efficacy and durability of these treatments are still under discussion [6,50], motivating the search for new biocompatible materials with permanent and fast effects associated with specific binding to the dentin surface, biocompatibility, and low irritability [6,7,50,51]. Here, we propose using strontium-containing carbonate nanoparticles that might exhibit all these properties.

We structurally characterized the SrCO_3_, Sr_0.5_Ca_0.5_CO_3_, and CaCO_3_ nanoparticles by X-ray diffraction, as depicted in Figure 1. We confirmed that pure CaCO_3_ (calcite) and SrCO_3_ (strontianite) minerals were formed by comparing the diffraction pattern of the nanoparticles with patterns extracted from the *Crystallography Open Database* (COD). We also indexed the mixed mineral Sr_0.5_Ca_0.5_CO_3_ based on the COD. Phase segregation was not evident. We assigned the shift of the diffraction peaks to higher 2θ values upon the addition of SrCO_3_ to substitution of Sr^2+^ for the lighter Ca^2+^ in the crystal lattice [52,53,54]. 

Particle size determined by DLS revealed diameters in the nanometric scale, which should favor penetration of the nanoparticles into the micro-sized dentin tubules [15,55]. The mean hydrodynamic diameter of SrCO_3_ nanoparticles was close to 387 nm (Table 1), resembling the diameter of the particles in Sensodyne^®^ Rapid Relief [15]. Moreover, the mean diameter of CaCO_3_ nanoparticles was 590 nm (Figure 2C). Small particles were present in the Sr_0.5_Ca_0.5_CO_3_ samples, which had hydrodynamic diameters centered at 66.7 nm (Table 1), thus guaranteeing their penetration into the dentin tubules. The polydispersity index (PDI) of the samples ranged from 0.6 to 0.9, revealing broad size distribution (Table 1). Although the PDI values were higher than those reported for homogeneous nanoparticle dispersions [15,55] (e.g., the PDI reported for the aqueous dispersion of the particles in Sensodyne^®^ Rapid Relief is 0.28 [15]), the whole size distribution was still at the nanometric scale, allowing the particles to be applied for dentin tubule obliteration.

Another important point regarding surface/particle interaction is charge. Since the dentin is negatively charged [56], positively charged or even neutral particles may be attracted to the surface by electrostatic or van der Waals forces, thus resulting in enhanced interaction. The zeta-potential (ζ) is influenced by the chemical composition of the particles and the dispersing medium dielectric constant, pH, and ionic strength [15,55]. The ζ values of SrCO_3_, CaCO_3,_ and Sr_0.5_Ca_0.5_CO_3_ nanoparticles were −2.08, −2.46, and −9.10 mV, respectively (Table 1). The ζ value increased upon substitution of Ca^2+^ for the less mobile Sr^2+^ in the mixed-carbonate particles. The low negative charge of all the particles should prevent repulsion and allow them to interact with the highly negative charge of the dentin surface [56]. Regarding the application of the nanoparticles as remineralizing and desensitizing agents with abrasive properties, their low ζ values allowed them to approach the dentin surface and deposit inside the dentin tubules [15,55]. The ζ value of Sensodyne^®^ Rapid Relief is −16.5 mV. Despite this higher ζ, this toothpaste still has positive effects when it comes to treating hypersensitivity [15]. 

Both SrCO_3_ nanoparticles and Sr_0.5_Ca_0.5_CO_3_ nanoparticles consisted of spherical nanoparticles (Figure 2A,B). In contrast, for CaCO_3_ nanoparticles, SEM revealed the formation of typical calcite orthorhombic micrometric structures (Figure 2C). Aggregation in the samples was a consequence of their low ζ [53]. The SEM images supported the hydrodynamic sizes found by DLS (Table 1). The larger size and orthorhombic morphology of CaCO_3_ nanoparticles partially explained their small effect on tubule obliteration and their ability to form continuous deposits on the dentin surface (Figure 6F). 

Going further, we investigated the cytotoxicity of the nanoparticles on hDPSCs cultures in vitro. Concentrations close to 10 µg/mL resulted in 20% in the cell viability compared to the control for the Sr_0,5_Ca_0,5_CO_3_ e SrCO_3_ nanoparticles (Figure 3a,b). This might indicate the interaction of the nanoparticles with the cell membrane, thus promoting a slight decrease in cell viability rather than a significant drop [57]. The ability of the nanoparticles to induce dentin remineralization has been investigated by their effect on TNAP activity [58]. The time-course events associated with TNAP activity have been previously reported [46,47]. Even though the TNAP messenger RNA transcription happens as early as the 7th day of cell culture, its highest activity occurs on the the14th day. Based on this, we sought to test both 7 days and 14 days. After 7 days of culture, the TNAP activity of odontoblast treated in the presence of CaCO_3_ nanoparticles, SrCO_3_ nanoparticles, and Sr_0.5_Ca_0.5_CO_3_ nanoparticles was like the control group (Figure 3C); however, on the day 14th, SrCO_3_ nanoparticles promoted increased TNAP activity compared to Sr_0,5_Ca_0,5_CO_3_ nanoparticles, although lower than the control group (Figure 3C). 

Several studies report the influence of different strontium concentration in TNAP activity, which corroborate our hypothesis. Bonnelye et al. demonstrated that TNAP gene expression and activity significantly increased when higher doses of strontium ranelate were provided [59]. Our group has also found increased TNAP activity on osteoblasts treated with a flavonoid-strontium complex [48]. Moreover, alizarin red staining (Figure 3D) showed that the presence of the particles significantly enhanced the mineral nodules deposition in comparison to the control group on the 14th day of culturing. The stimulus upon mineralization was more significant for CaCO_3_ nanoparticles and Sr_0.5_Ca_0.5_CO_3_ nanoparticles, which might be related to the Ca^2+^ ions already present in the particles’ structure. The formation of mineralized nodules indicates the differentiation of hDPSCs in an osteogenic phenotype [40]. Moreover, the deposition of minerals is an indicator of matrix vesicle secretion by the cells, which usually occurs when the cells switch from an immature to a mature phenotype [60]. These results were confirmed by the analysis of the expression of the main osteogenic markers COL1, ALP, and RUNX2. All the particles stimulated the expression of COL1 and ALP and sustained the expression of the transcriptional factor RUNX2-related genes after 7 days of culturing, compared to the control. These genes are earlier markers of differentiation of hDPSCs to an osteoblastic phenotype [61,62,63], which supports the enhanced mineral nodule formation after treatment with the particles (Figure 3D). SrCO_3_ nanoparticles stimulated the expression of the genes related to these three osteogenic markers even after 14 days of culturing (Figure 4). 

After conducting biological assays in vitro, we mixed the SrCO_3_, Sr_0.5_Ca_0.5_CO_3_, or CaCO_3_ nanoparticles with PVA to obtain gel-like samples, which we further used to brush the dentin disc surface. The ATR-FTIR spectra of the dentin samples before and after brushing helped to investigate changes in the chemical composition of the dentin surface after contact with the nanoparticles. Figure 5A illustrates the spectra of the dentin discs brushed with SrCO_3_ nanoparticles. Bands assigned to biological apatite emerged in the 1100–1000 cm^−1^ range and were related to asymmetric stretching of the phosphate group [53,64,65]. The typical amide III band of collagen also appeared at 1250 cm^−1^, attesting to the presence of this structural protein on the dentin surface [66,67]. Bands ascribed to carboxylate and amine groups were observed between 1750 and 1500 cm^−1^ [68]. The natural substitution of phosphate with carbonate groups is a fingerprint of biological apatite [53,64,65,69,70,71]. However, the intensity of the bands at 1480 and 850 cm^−1^ increased after brushing with SrCO_3_ (Figure 5A—red line and black arrows), while there was no evidence of changes in the band at 1070 cm^−1^ attributed to phosphate. The intensity of carbonate-related bands increased as a function of the number of brushing cycles (Figure 5A—green and blue lines). This result supported the accumulation of nanoparticles due to sequential brushing, reinforcing the ability of SrCO_3_ nanoparticles to bind to the dentin surface without modifying the apatite mineral structure. 

The samples brushed with Sr_0.5_Ca_0.5_CO_3_ nanoparticles behaved similarly: the bands related to carbonate groups intensified upon sequential brushing cycles for 7 (Figure 5B—green line) and 14 days (Figure 5B—blue line). We then compared the effects of SrCO_3_ nanoparticles and Sr_0.5_Ca_0.5_CO_3_ nanoparticles: after the first brushing cycle, the application of SrCO_3_ nanoparticles inverted the relative intensity between the bands at 1070 cm^−1^, assigned to phosphate, and 850 cm^−1^, assigned to carbonate, whereas brushing with Sr_0.5_Ca_0.5_CO_3_ nanoparticles elicited this inversion only after 7 days_._ We attributed this result to the higher negative ζ value of Sr_0.5_Ca_0.5_CO_3_ nanoparticles, which translated into lower ability to bind to the dentin surface as compared to SrCO_3_ nanoparticles. The fact that the phosphate band slightly shifted to lower wavenumbers after brushing with Sr_0.5_Ca_0.5_CO_3_ was noteworthy and indicated that Ca^2+^ was replaced with the heavier Sr^2+^ in the apatite lattice [53,65,69]. The intensity of the band at 1480 cm^−1^ (black arrow) did not change significantly after brushing with CaCO_3_ nanoparticles, which indicated a lower deposition of the particles on the dentin surface. Given that SrCO_3_ and CaCO_3_ nanoparticles have similar ζ values, this result attested to the affinity of Sr^2+^ for the dentin surface and the enhanced binding ability of the strontium-containing carbonate nanoparticles. Moreover, the larger size and orthorhombic morphology of CaCO_3_ nanoparticles may have prevented their contact with the dentin surface and penetration into the dentin tubules, as also suggested by SEM analysis (Figure 6F). The higher intensity of the phosphate band after brushing with CaCO_3_ nanoparticles for 14 days could result from higher exposure of the apatite mineral, assigned to the removal of the smear layer due to the well-known abrasive effect of this compound [2,31]. Similar to the effect of the strontium-containing carbonate nanoparticles prepared herein, brushing with Sensodyne^®^ Rapid Relief (strontium acetate as an active ingredient) intensified the band attributed to phosphate and shifted it to a lower wavenumber. Nevertheless, the bands related to carbonate remained unchanged. 

We calculated tubule obliteration by Equation (1) by counting the total number of tubules and classifying them into the open, partially obliterated, and obliterated in a 10 mm^2^ area of an SEM image. The results presented in Table 2 evidenced the low obliteration ability (24.7%) of CaCO_3_. This finding supported the absence of changes in the bands associated with carbonate in the FTIR spectrum obtained for dentin after brushing with CaCO_3_ nanoparticles. Nevertheless, the spherical strontium-containing nanoparticles promoted higher obliteration (67.7%) for the samples brushed with SrCO_3_ nanoparticles, which remained stable after acidic treatment (Table 2). The samples brushed with Sr_0.5_Ca_0.5_CO_3_ nanoparticles had the highest obliteration values, which were even higher after the acidic challenge when they reached 100% (Table 2). This could be due to the higher solubility of CaCO_3_ nanoparticles in an acidic medium, which led to the re-deposition of smaller particles that were able to penetrate into the dentin tubules as a result of a well-known dissolution/reprecipitation process [72,73]. Surprisingly, although brushing with Sensodyne^®^ Rapid Relief promoted intertubular deposition of the particles, this toothpaste presented the lowest obliteration ability, which was close to 19.5% before and 31.9% after acidic treatment. The side-view SEM images evidenced changes in the morphology of the internal tubule walls and the penetration of particles (Figure 7). In particular, the red arrow in Figure 6 indicates the presence of nanoparticles inside the tubules of the samples brushed with Sr_0.5_Ca_0.5_CO_3_ nanoparticles, evidencing the high affinity of these nanoparticles for the dentin structure.

We carried out elemental analysis of the dentin surface before and after brushing by EDS coupled with SEM. The relative amount of Ca, Sr, and P depicted in Table 3 revealed a (Ca + Sr)/P molar ratio close to 1.8 for clean dentin samples before brushing. The stoichiometric ratio expected for hydroxyapatite is 1.67. The higher values could be correlated to cationic and anionic substitutions in the biological apatite structure. The FTIR data supported this finding: bands related to B-type substitution, in which phosphate groups are replaced with carbonate, were present in the spectra [24,74]. Furthermore, the presence of a small amount of strontium before brushing was related to the natural substitution of Ca^2+^ for Sr^2+^ in mineralized tissue. Strontium uptake at concentrations close to 2.4 mg/day from water and diet has been reported [2,23,24,25,74]. Brushing with CaCO_3_ nanoparticles reduced the (Ca + Sr)/P molar ratio due to exposure of phosphate because of abrasion, as also evidenced by FTIR analysis. After the acid attack, the relative amount of Ca decreased, indicating partial dentin demineralization [2,23,24,25]. The (Ca + Sr)/P molar ratio after brushing with SrCO_3_ nanoparticles was like the ratio in clean dentin (Table 3). However, the relative strontium content increased, indicating that strontium deposited on the dentin surface. We identified homogeneous strontium deposits in the EDS images in Figure 8. After the acid attack, the relative amount of calcium decreased, which translated into increased strontium and phosphorus content (Figure 9 and Table 3). Calcium removal and maintenance of strontium deposited on the dentin surface were evident in the EDS images (Figure 9). The lack of a green color, related to calcium, and the presence of a blue color, related to strontium, in the side view of Figure 9 after the acid attack supported the higher affinity of strontium for the dentin surface, which could result in remineralization and obliteration [69,75]. We verified a similar behavior after we brushed dentin discs with the mixed Sr_0.5_Ca_0.5_CO_3_ nanoparticles: calcium and strontium were homogeneously deposited (Table 3, Figure 8), and calcium was partially removed after the acid attack (Figure 9). Calcium removal was less accentuated as compared to the samples brushed with SrCO_3_ nanoparticles, which indicated that the coating formed by deposition of Sr_0.5_Ca_0.5_CO_3_ nanoparticles on dentin was more resistant to acid attack (Figure 9). The presence of green spots in the side view images indicated that higher amounts of calcium remained deposited on the dentin surface after acid attack as compared to the samples brushed with SrCO_3_ nanoparticles. The reduction in the phosphorus content (Table 3 and Figure 9) could be assigned to the formation of carbonated apatite as supported by the FTIR data [76]. Brushing with Sensodyne^®^ Rapid Relief resulted in a higher amount of strontium in dentin as compared to the samples brushed with the tested nanoparticles (Table 3). Nevertheless, intertubular despite intratubular deposition was evident from the presence of the black holes observed in the EDS image attributed to non-obliterated tubules (Figure 8). After the acid attack, a reduced amount of strontium remained on the dentin samples brushed with Sensodyne^®^ Rapid Relief as compared to the samples brushed with SrCO_3_ nanoparticles or Sr_0.5_Ca_0.5_CO_3_ nanoparticles (Table 3), revealing the lower resistance of the coating formed by the commercial toothpaste as compared to the coating formed by the new active ingredients containing strontium proposed herein. 

## 5. Conclusions

Dentin samples brushed with the strontium-containing nanoparticles were coated with a mineral layer that was also able to penetrate dentin tubules, while CaCO_3_ nanoparticles remained as individual particles on the dentin surface. Moreover, after brushing with SrCO_3_ nanoparticles or Sr_0.5_Ca_0.5_CO_3_ nanoparticles, the bands related to carbonate groups intensified. The slight shift of the phosphate-related FTIR band to the lower wavenumber indicated that strontium replaced calcium on the dentin structure after treatment. The coating promoted by SrCO_3_ nanoparticles or Sr_0.5_Ca_0.5_CO_3_ nanoparticles resisted an acidic environment, while calcium and phosphorus were removed from the top of the dentin surface. The nanoparticles were not toxic to hDPSCs, sustained TNAP activity, and promoted mineralization. The performance of the strontium-containing nanoparticles in terms of tubule obliteration and resistance to acid attack was even better as compared to Sensodyne^®^ Rapid Relief. Adding Sr_0.5_Ca_0.5_CO_3_ nanoparticles as an active ingredient in dentifrice formulations may be commercially advantageous because this compound combines the well-known abrasive properties of calcium carbonate with the mineralization ability of strontium, while the final cost remains between the costs of CaCO_3_ nanoparticles and SrCO_3_ nanoparticles.

## Figures and Tables

**Figure 1 jfb-13-00250-f001:**
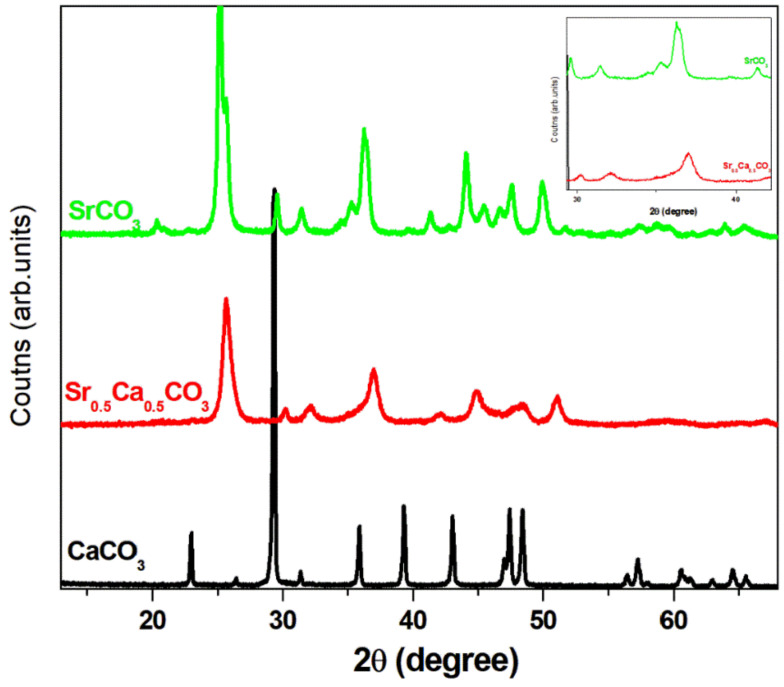
**Structural characterization of the particles**. X-ray diffraction patterns of SrCO_3_ (green line), Sr_0.5_Ca_0.5_CO_3_ (red line), and CaCO_3_ (black line). The structures correspond to CaCO_3_ calcite (COD-1010962), SrCO_3_ strontianite (COD-5000093), and a mixed Sr_0.5_Ca_0.5_CO_3_ mineral (COD 9015600). The displacement of the peaks to higher 2θ in the case of Sr_0.5_Ca_0.5_CO_3_ as compared to SrCO_3_ evidences the presence of a lighter element in the structure (Ca) (see image inserted in the figure).

**Figure 2 jfb-13-00250-f002:**
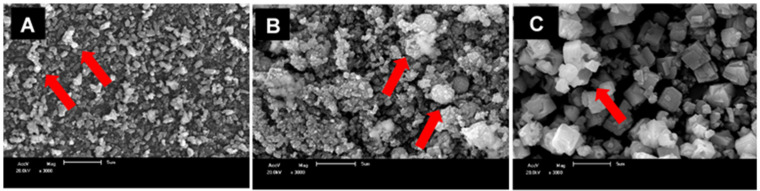
SEM images of the SrCO_3_ (**A**), Sr_0.5_Ca_0.5_CO_3_ (**B**), and CaCO_3_ (**C**) nanoparticles. The red arrows indicate the presence of micrometric particles due to aggregated nanosized particles. The scale bar corresponds to 5 µm. The images were acquired using accelerating voltage of 20 kV and 3000× magnification.

**Figure 3 jfb-13-00250-f003:**
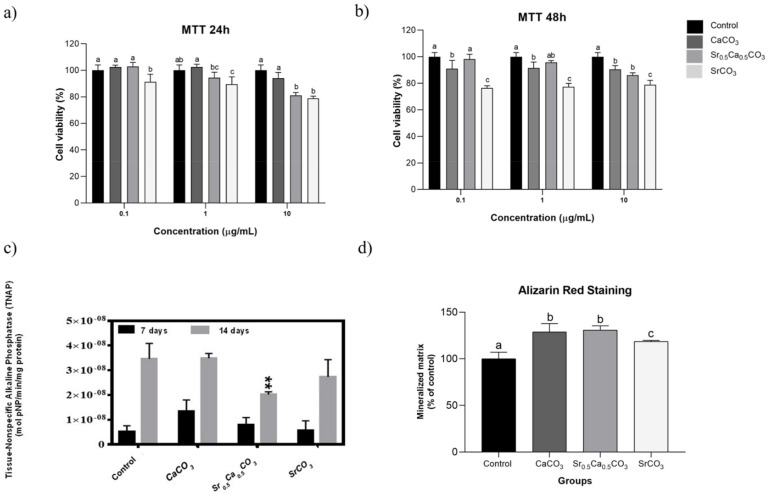
**Effect of the nanoparticles on human dental stem cells (hDPSCs) in vitro.** Cell viability of hDPSCs after 24 h (**a**) and 48 h (**b**) of culturing the presence of the nanoparticles measured by MTT assay. The results represent mean ± SD of five experiments. Statistical analysis was carried out by two-way ANOVA, and Tukey’s post-test were performed to compare the results. Bars with different letters represent significant differences between groups for each concentration of particles tested. Tissue-Non-specific Alkaline Phosphatase (TNAP) activity after 7 and 14 days of culturing in the presence and absence (control) of 1 µg/mL of particles (**c**). For the determination of TNAP activity, multiple statistical comparisons were performed by one-way ANOVA in comparison with the control, ** = 0.0014. The results represent mean ± SD of three experiments. Evaluation of the formation of the mineralized nodules by alizarin red staining after 14 days of culturing in the presence and absence (control) of 1 µg/mL of particles (**d**). The results represent mean ± SD of five experiments. Statistical analysis was carried out by two-way ANOVA, and Tukey’s post-test was performed to compare the results. Bars with different letters represent significant differences between groups. All the controls correspond to hDPSC cultured on the polystyrene plate in absence of the nanoparticles.

**Figure 4 jfb-13-00250-f004:**
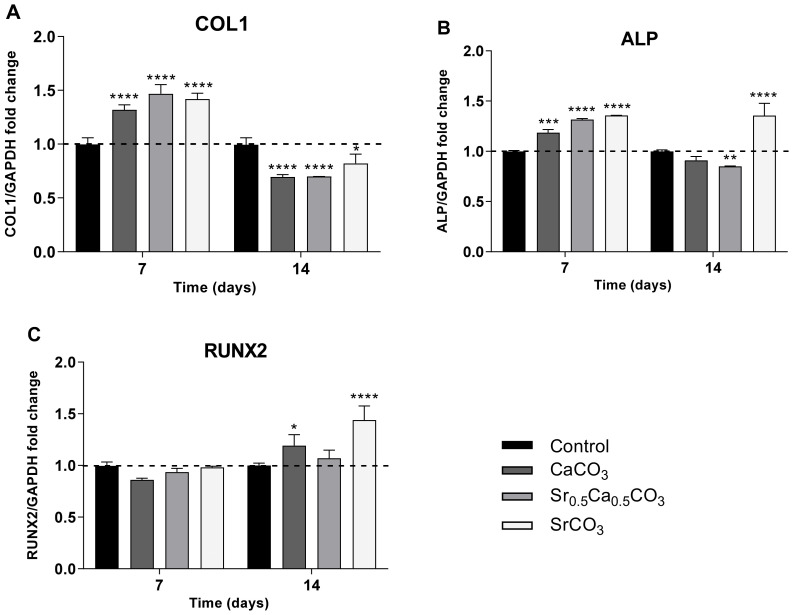
Expression of the osteogenic markers COL1 (**A**), ALP (**B**), and RUNX2 (**C**) by hDPSCs cultivated in the presence of 1 µg/mL Sr-carbonates for 7 and 14 days. The control corresponds to the cells cultivated in polystyrene plate in absence of the particles. Statistical analyses were carried out by two-way ANOVA, and Tukey’s post-test was performed in relation to the control; **** *p* < 0.0001, *** *p* < 0.001, ** *p* < 0.01, and * *p* < 0.05.

**Figure 5 jfb-13-00250-f005:**
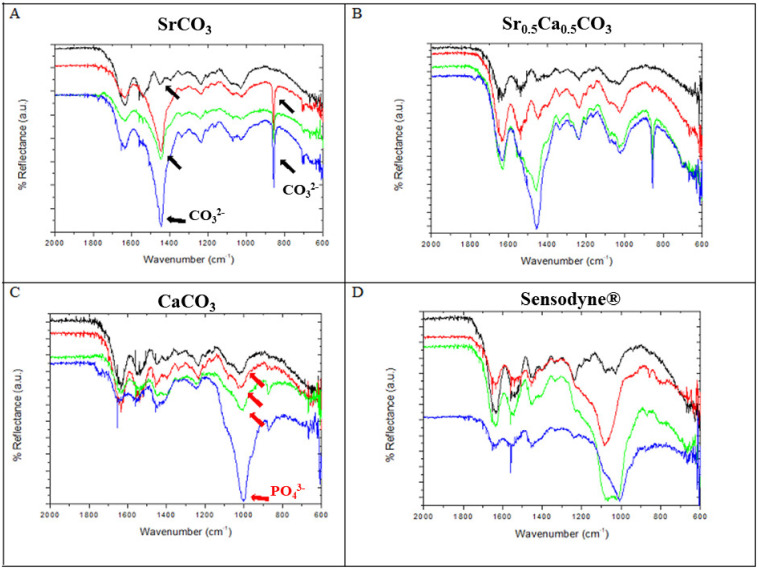
**Chemical analysis of the dentin discs before and after brushing.** ATR-FTIR of the dentin surface before (black lines) and after brushing for 1 (red lines), 7 (green lines), and 14 (blue lines) days with SrCO_3_ nanoparticles (**A**), Sr_0.5_Ca_0.5_CO_3_ nanoparticles (**B**), CaCO_3_ nanoparticles (**C**), and Sensodyne^®^ Rapid Relief (**D**). The black and red arrows indicate changes in the bands assigned to the presence of CO_3_^2−^ and PO_4_^3−^ groups, respectively, after brushing.

**Figure 6 jfb-13-00250-f006:**
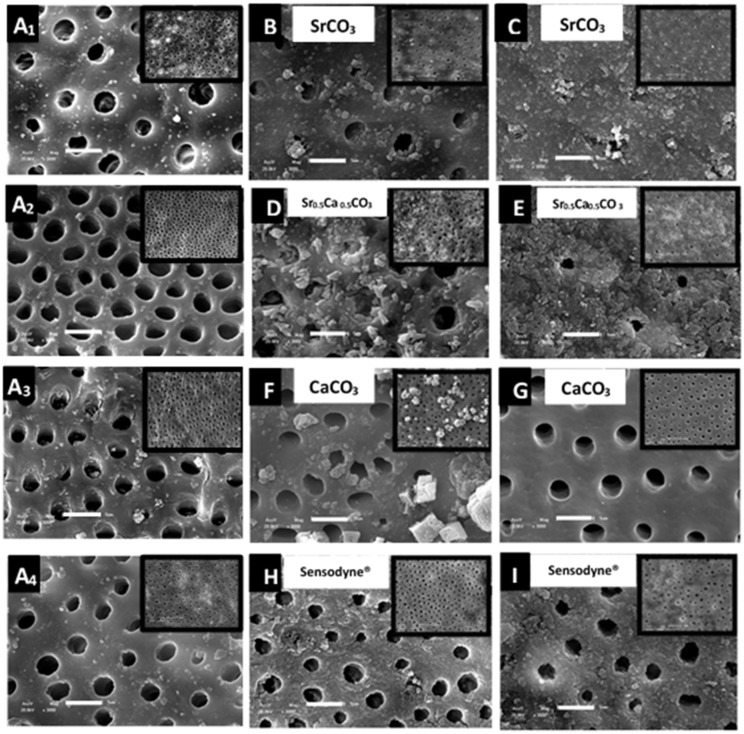
SEM imagens of the dentin specimens before (**A_1_**–**A_4_**) and after brushing with SrCO_3_ nanoparticles (**B**,**C**), Sr_0.5_Ca_0.5_CO_3_ nanoparticles (**D**,**E**), CaCO_3_ nanoparticles (**F**,**G**), and Sensodyne^®^ Rapid Relief (**H**,**I**). Images C, E, G, and I were acquired after acid challenge. The inserts correspond to images with low amplification. The scale bar corresponds to 5 µm.

**Figure 7 jfb-13-00250-f007:**
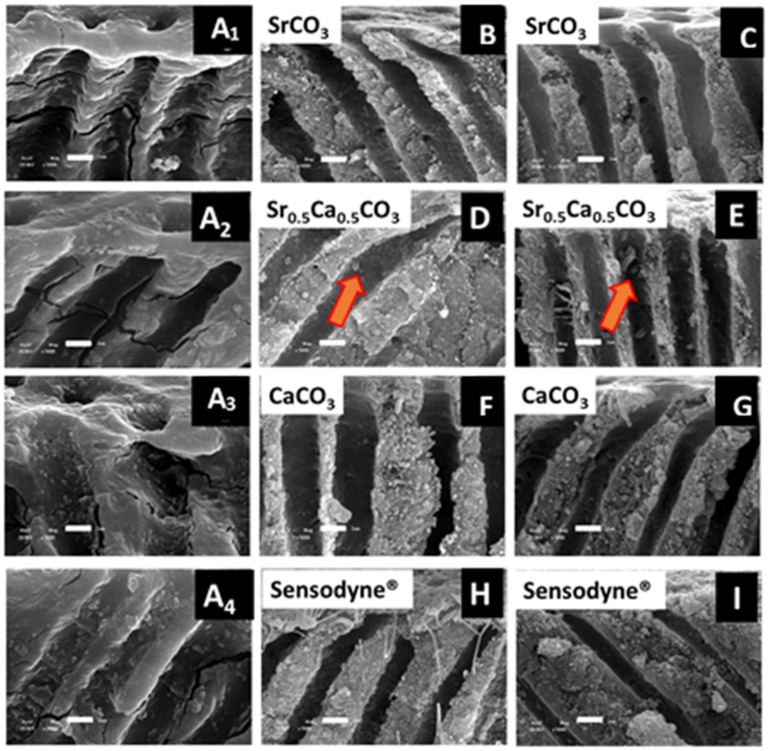
Transverse view SEM imagens of the dentin specimens before (**A_1_**–**A_4_**) after brushing with SrCO_3_ nanoparticles (**B**,**C**), Sr_0.5_Ca_0.5_CO_3_ nanoparticles (**D**,**E**), CaCO_3_ nanoparticles (**F**,**G**) and Sensodyne^®^ Rapid Relief (**H**,**I**). Images C, E, G, and I were acquired after acid challenge. The scale bar corresponds to 2 µm. The orange arrows indicate the presence of particles inside the dentin tubules.

**Figure 8 jfb-13-00250-f008:**
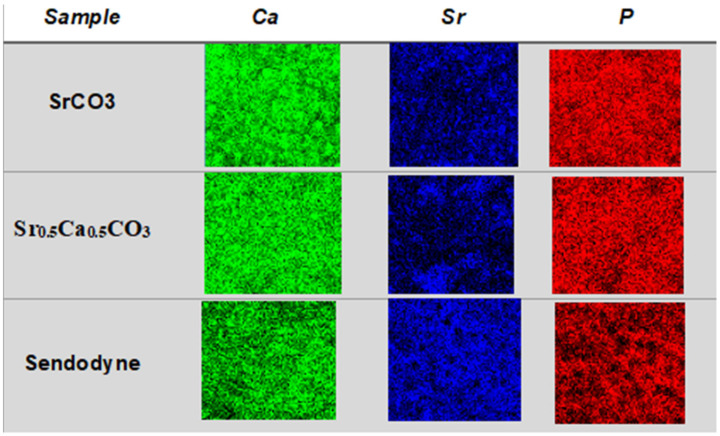
**Calcium (Ca), Strontium (Sr), and Phosphorous (P) deposition on the surfaces.** EDS maps of Ca (green), Sr (dark blue), and P (red) present on the dentin surface (top view) after brushing with strontium-containing carbonate nanoparticles.

**Figure 9 jfb-13-00250-f009:**
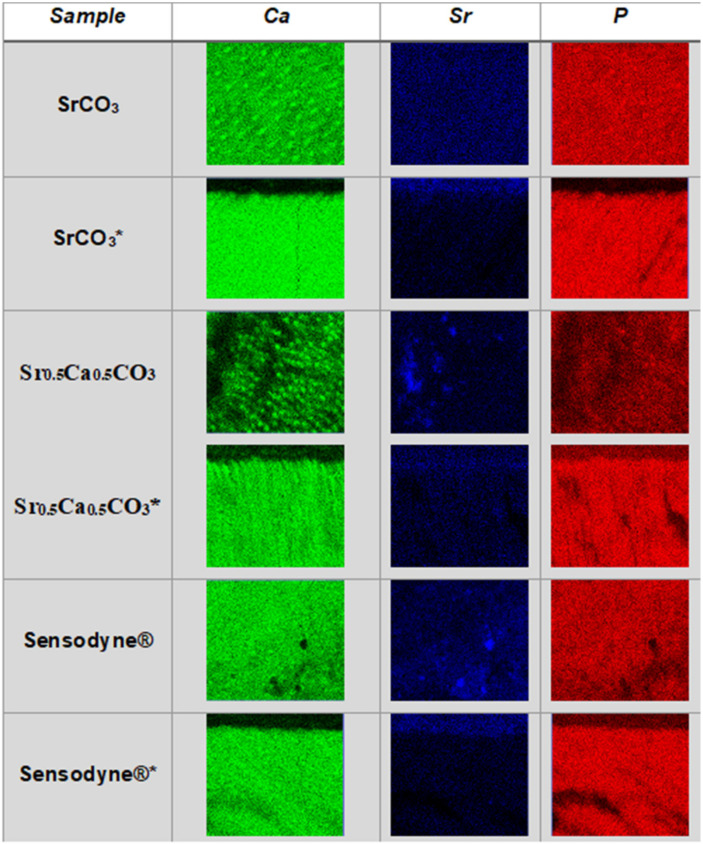
**Resistance of the coatings to the acidic treatment.** EDS maps of Ca (green), Sr (dark blue), and P (red) present at the dentin top and side view after brushing with Sr-containing nanoparticles and after acid attack. * Indicates the images obtained from the transversal cut of the samples.

**Table 1 jfb-13-00250-t001:** Hydrodynamic diameter, polydyspersity index (PDI), and zeta-potential (ζ) determined by DLS. The number between parenthesis in the diameter column corresponds to the full width at the half maximum of the size distribution.

Sample	Diameter (nm)	PDI	ζ (mV)
**SrCO_3_**	387.0 (166)	0.56	−2.1 ± 3.2
**Sr_0.5_Ca_0.5_CO_3_**	66.7 (16.7)	0.58	−9.1 ± 4.5
**CaCO_3_**	590.8 (156.5)	0.90	−2.5 ± 4.8

**Table 2 jfb-13-00250-t002:** Percentage of obliterated dentin tubules (*n*) on the surface of the specimens after brushing with gel containing nanoparticles, followed (+) or not (−) by acid attack (AA).

Sample	AA	*n* (%)
**Control**	**−**	0% ± 0.0
**SrCO_3_** **nanoparticles**	**−**	67.7% ± 0.01
	**+**	65.7% ± 0.08
**Sr_0.5_Ca_0.5_CO_3_ nanoparticles**	**−**	93.3% ± 0.09
	**+**	100% ± 0.0
**CaCO_3_** **nanoparticles**	**−**	24.7% ± 0.10
	**+**	0% ± 0.0
**Sendodyne^®^**	**−**	19.5% ± 0.08
	**+**	31.9% ± 0.15

**Table 3 jfb-13-00250-t003:** Quantification by EDS of the chemical elements present on the surface of dentin specimens before (control) and after brushing with gel containing nanoparticles, followed (+) or not (−) by acid attack (AA).

Sample	AA	Ca	Sr	P	(Ca + Sr)/P
**Control**	−	63.47	1.00	35.52	1.81 ± 0.17
**SrCO_3_** **nanoparticles**	−	59.20	6.00	34.90	1.87 ± 0.17
	**+**	24.90	26.92	48.18	1.07 ± 0.17
**Sr_0.5_Ca_0.5_CO_3_ nanoparticles**	−	59.67	4.54	35.78	1.79 ± 0.17
	**+**	43.07	47.03	9.90	9.10 ± 0.17
**CaCO_3_** **nanoparticles**	−	59.50	0.90	39.61	1.52 ± 0.17
	**+**	53.30	4.36	42.33	1.36 ± 0.17
**Sendodyne^®^**	−	40.58	16.67	42.75	1.34 ± 0.17
	**+**	50.49	5.56	43.95	1.27 ± 0.17

## Data Availability

The data presented in this study are available on request from the corresponding author.

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
