# Peer review of "Strontium Carbonate and Strontium-Substituted Calcium Carbonate Nanoparticles Form Protective Deposits on Dentin Surface and Enhance Human Dental Pulp Stem Cells Mineralization"

_jfb, 2022, doi:10.3390/jfb13040250_

Round 1
Reviewer 1 Report
In this study, the authors demonstrate a comprehensive characterization and assessment of strontium carbonate as a potential novel material for treatment of tooth sensitivity. This is a very interesting and clinically-relevant study. The experiments are design in a logical way – they properly address the characteristics that are relevant for the potential use of the material in dentistry. The material is characterized in comparison with the currently used treatments, which increases the potential impact of the findings. The potential advantage of strontium carbonate over currently used materials is that it combines tooth-protective and abrasive properties. This is a very valuable study, especially now when commonly used fluoride is under the scrutiny for its potential toxic effects and there is a need for development of new materials protecting teeth. Overall, this is well written manuscript and well design study – I have no major critiques, neither any suggestions for additional experiments.
Points to address in the revised manuscript:
1. In Materials – add statement about the approval of the study of teeth obtained from patients.
2.In Methods – add a section describing statistical analyses and a section describing gene expression analyses.
3. In results: indicate in which experiment cells were grown in osteogenic medium – this is not clear now.
4. The description of Figure 1 in the results section and in the legend is not clear. It is hard to figure out which peak represents what. Please clarify.
5. In the Results, please provide rationale for analyzing nanoparticles size and charge (this is explained later in the discussion, but explaining the purpose of analyses before showing data puts data in the context – easier to appreciate the results).
6. Please clarify the results of the statistical analyses shown on figure 3 – it is not clear what the letters mean and what were the comparison groups, and where are statistically significant differences.
7. I suggest combining figure 7 with figure 6, since they are showing complementary analyses – shown together would highlight better the superiority of the strontium materials over the currently used ones. This is such an excellent set of data.
8. Section 3.4 is hard to follow – it seems that the description of the results does not match the data shown in Table3 that shows more Ca loss in SrCO3 group – please clarify.
Other (minor) edits/corrections:
1. Add reference (line 87) to the statement about the study showing effect of strontium on osteointegration of implants.
2. Edit (line 99): remove either “including” or “like”.
3. Correct (line 122): should be cementoenamel (not amelocemental).
4. Clarify or change wording (line 209): “images helped to quantify”.
5. Add (line 264) after X-ray “diffraction”.
6. Add (line326) “enzyme” between “main” and “responsible”.
7. Correct (line 344) should be transcription not transcriptional.
8. Please label images on Figure 5 with what each panel shows and what the peaks of interest – it would be easier to understand this figure at glance.
9. I suggest to replace (line 420) the “side- view” with “transverse view.
10. lines 500&501: by saying “avoid repulsion” did the authors mean “attract”?
11. line520: I suggest replacing “promoted a reduction” with “resulted in“.
12. Line 526: I suggest replacing shorter” with “early”.
13.Line 533: delete “higher”, since later in the sentence the authors say “increased”
Author Response
Dear Reviewer,
We would like to sincerely thank you for the comments on our manuscript, which were very useful for improving the quality of the text. We have taken all the comments into account in the revised version of the paper. All the changes are highlighted in main the text.
Reviewer 1
In this study, the authors demonstrate a comprehensive characterization and assessment of strontium carbonate as a potential novel material for treatment of tooth sensitivity. This is a very interesting and clinically-relevant study. The experiments are design in a logical way – they properly address the characteristics that are relevant for the potential use of the material in dentistry. The material is characterized in comparison with the currently used treatments, which increases the potential impact of the findings. The potential advantage of strontium carbonate over currently used materials is that it combines tooth-protective and abrasive properties. This is a very valuable study, especially now when commonly used fluoride is under the scrutiny for its potential toxic effects and there is a need for development of new materials protecting teeth. Overall, this is well written manuscript and well design study – I have no major critiques, neither any suggestions for additional experiments.
Points to address in the revised manuscript:
- Rev #1: In Materials – add statement about the approval of the study of teeth obtained from patients.
Answer: Thank you very much for your comment. The statement about the approval of the study of teeth obtained from patients is described in the “Materials” section as follow: “Dentin discs were prepared from third molar teeth from the tooth bank of the Dental School (University of São Paulo, Ribeirão Preto, Brazil) after approval of the Ethics in Research Committee of the same institute (06245218.2.0000.5419).”
- In Methods – add a section describing statistical analyses and a section describing gene expression analyses.
Answer: Thank you very much for pointing this out. The description of the gene expression analyses was added to the methodology (section 2.2.13). Since different statistical analyses were used for different data, the description was added to the legend of each figure.
- In results: indicate in which experiment cells were grown in osteogenic medium – this is not clear now.
Answer: all the cells culture was carried out in ostegenic medium. A sentence was added to the methodology to make it clear.
- The description of Figure 1 in the results section and in the legend is not clear. It is hard to figure out which peak represents what. Please clarify.
Answer: We agree that the description was not helpful. The figure was changed, and the legend modified, as suggested.
- In the Results, please provide rationale for analyzing nanoparticles size and charge (this is explained later in the discussion, but explaining the purpose of analyses before showing data puts data in the context – easier to appreciate the results).
Answer: We thank the reviewer for this suggestion. A couple of sentences was added to the size and zeta potential results section.
- Please clarify the results of the statistical analyses shown on figure 3 – it is not clear what the letters mean and what were the comparison groups, and where are statistically significant differences.
Answer: We thank the reviewer for the suggestion. The legend of the figure was changed in order to make clear the statistical analysis.
- I suggest combining figure 7 with figure 6, since they are showing complementary analyses – shown together would highlight better the superiority of the strontium materials over the currently used ones. This is such an excellent set of data.
Answer: We thank the reviewer for the suggestion and for finding the data interesting. However, our attempt to merge the figures resulted in very small carts, which loose the details. In this sense, we would prefer to leave the figures are they are.
- Section 3.4 is hard to follow – it seems that the description of the results does not match the data shown in Table3 that shows more Ca loss in SrCO3 group – please clarify.
Answer: We thank the reviewer for this comment. Some sentences of the section 3.4 were re-written. The important point here to consider is that the calculations of strontium, calcium and phosphorous content is relative. Another point is the possible contribution of the elemental content in the depth to the analysis of the top view images.
Other (minor) edits/corrections:
- Add reference (line 87) to the statement about the study showing effect of strontium on osteointegration of implants.
Answer: Thank you very much for your comment. For a better understanding and improvement of the work, we added reference in this sentence
- Edit (line 99): remove either “including” or “like”.
Answer: Thank you very much for your comment. We removed the word “like”
- Correct (line 122): should be cementoenamel (not amelocemental).
Answer: Thank you very much for your comment. The correct terminology was added to the text.
- Clarify or change wording (line 209): “images helped to quantify”.
Answer: We thank the reviewer for the comment. The sentence was changed.
- Add (line 264) after X-ray “diffraction”.
Answer: Thank you very much for your comment. We added the word “diffraction” after X-ray.
- Add (line326) “enzyme” between “main” and “responsible”.
Answer: Thank you very much for your comment. The word “enzyme” was added to the text.
- Correct (line 344) should be transcription not transcriptional.
Answer: Thank you very much for your point this out. It was corrected in the text.
- Please label images on Figure 5 with what each panel shows and what the peaks of interest – it would be easier to understand this figure at glance.
Answer: We fully agreed with the suggestion. The figure as changed as suggested.
- I suggest to replace (line 420) the “side- view” with “transverse view.
Answer: Thank you very much for your comment. The words were replaced.
- lines 500&501: by saying “avoid repulsion” did the authors mean “attract”?
Answer: We thank the reviewer for the comment. Positively charged particles can be attracted to dentin surface by electrostatic interaction, and the neutral particles can be attracted by van der Waals forces. The sentence was changed.
- line520: I suggest replacing “promoted a reduction” with “resulted in“.
Answer: The sentence was changed accordingly.
- Line 526: I suggest replacing shorter” with “early”.
Answer: The sentence was changed accordingly.
- Line 533: delete “higher”, since later in the sentence the authors say “increased”
Answer: The sentence was changed accordingly.

Reviewer 2 Report
Hello,
I consider the study "Strontium carbonate and strontium-substituted calcium carbonate nanoparticles form protective deposits on dentin surface and enhances human dental pump stem cells mineralization" very interesting in terms of the topic as well as the complexity of the work method, but the study requires changes to improve the understanding of the results obtained.
· Title of the article: I recommend rewriting the title because it has too little information about the top study and the techniques used. To replace the word "pump" with "pulp" (line 2).
· Abstract: Rewriting the abstract following the technical editing criteria (aim, material and methods...)
· Keywords in alphabetic order
· Introduction: Try to choose another term for ‘’exploited’’(line 94)
· Aim: A clear formulation of the purpose of the study, which must also be found in the abstract(lines 101-107)
· Material and methods: Well presented. I recommend introducing the Statistical Analysis section(line 257)
· Results: Well presented……Line 280, table 1 – PI or PDI??
· Discussion: The discussions must be reformulated. The references made to the results (numbers, tables, graphs, images) must be described in the results section. Try to avoid repeating the description of the results in the discussions.
· Conclusion: The conclusions must be reformulated with direct and concise references to the effects of tested materials.
Author Response
Dear Reviewer,
We would like to sincerely thank you for the comments on our manuscript, which were very useful for improving the quality of the text. We have taken all the comments into account in the revised version of the paper. All the changes are highlighted in main the text.
Reviewer 2
I consider the study "Strontium carbonate and strontium-substituted calcium carbonate nanoparticles form protective deposits on dentin surface and enhances human dental pump stem cells mineralization" very interesting in terms of the topic as well as the complexity of the work method, but the study requires changes to improve the understanding of the results obtained.
Rev #2: Title of the article: I recommend rewriting the title because it has too little information about the top study and the techniques used. To replace the word "pump" with "pulp" (line 2).
Answer: We thank the reviewer for the suggestion and for pointing this mistake out. The word pump was replaced by pulp in the tittle.
Rev #2: Abstract: Rewriting the abstract following the technical editing criteria (aim, material and methods...)
Answer: We thank the reviewer for the comment. The abstract was written following the instruction for author found in the journal submission system, as follow: “The abstract should be a total of about 200 words maximum. The abstract should be a single paragraph and should follow the style of structured abstracts, but without headings: 1) Background: Place the question addressed in a broad context and highlight the purpose of the study; 2) Methods: Describe briefly the main methods or treatments applied. Include any relevant preregistration numbers, and species and strains of any animals used. 3) Results: Summarize the article's main findings; and 4) Conclusion: Indicate the main conclusions or interpretations. The abstract should be an objective representation of the article: it must not contain results which are not presented and substantiated in the main text and should not exaggerate the main conclusions.” A sentence containing the aim of the study was added to the abstract to make it clear.
Rev #2: Keywords in alphabetic order
Answer: We thank the reviewer for the comment. The keywords are now in alphabetic order.
Rev #2: Introduction: Try to choose another term for ‘’exploited’’(line 94)
Answer: We thank the reviewer for the comment. The word “exploited” was relaced by “studied”.
Rev #2: Aim: A clear formulation of the purpose of the study, which must also be found in the abstract (lines 101-107)
Answer: We thank the reviewer for the comment. The final of the introduction was changed to make the aim of the study clear.
Rev #2: Material and methods: Well presented. I recommend introducing the Statistical Analysis section (line 257)
Answer: We thank the reviewer for the comment. Since different statistical analysis were performed for different experiments, a sentence about the data was added at the end of the figures´ legend.
Rev #2: Results: Well presented……Line 280, table 1 – PI or PDI??
Answer: We thank the reviewer for point this out. The correct is PDI. It was changed in the table.
Rev #2: Discussion: The discussions must be reformulated. The references made to the results (numbers, tables, graphs, images) must be described in the results section. Try to avoid repeating the description of the results in the discussions.
Answer: We thank very much the reviewer for this comment. The references made to the results in the discussion was only to help and to guide the reader through the discussion of the data presented in the previous section. We tried to avoid repetitions. The separation of the sections in results and then discussion is also an instruction for the authors.
Rev #2: Conclusion: The conclusions must be reformulated with direct and concise references to the effects of tested materials
Answer: We thank the reviewer for this comment. Some sentences were cut from the conclusions to make it more concise and direct.

Reviewer 3 Report
1. Previous studies mentioned that there is only minimal evidence for the efficacy of both strontium- and potassium-based toothpastes in relieving symptoms of dentine hypersensitivity DH. can you please justify your conclusion in the light of this fact. (you need to discuss this point in introduction and discussion sections)
2.the following study support the use of an aluminium lactate/potassium nitrate/hydroxylapatite toothpaste for DH management. please compare your results with the findings of this study.
Seong J, Newcombe RG, Foskett HL, Davies M, West NX. A randomised controlled trial to compare the efficacy of an aluminium lactate/potassium nitrate/hydroxylapatite toothpaste with a control toothpaste for the prevention of dentine hypersensitivity. J Dent. 2021 May;108:103619. doi: 10.1016/j.jdent.2021.103619. Epub 2021 Feb 26. PMID: 33647373.
Author Response
Dear Reviewer,
We would like to sincerely thank you for the comments on our manuscript, which were very useful for improving the quality of the text. We have taken all the comments into account in the revised version of the paper. All the changes are highlighted in main the text.
Reviewer 3
- Rev #3: Previous studies mentioned that there is only minimal evidence for the efficacy of both strontium- and potassium-based toothpastes in relieving symptoms of dentine hypersensitivity DH. can you please justify your conclusion in the light of this fact (you need to discuss this point in introduction and discussion sections)
Answer: It is a very important point. The use of strontium in the form of an acetate or other high soluble salts, may not the efficient for the delivery and enhancement of the deposition of strontium on the surface and inside the wall of dentin tubules, which might be the reason of the low evidence of efficacy. For this reason, we decided to synthesize carbonate nanoparticles (less soluble) and to compare the effect with Sendodyne Rapid Relief (based on strontium acetate). For instance, the effect of the strontium on calcium replacement at the structure of hydroxyapatite and on the stimulus of cell mineralization have been reported by our group (see REFS 48, 52, and 69), which may also help in remineralization of dentin, relieving symptoms of dentin hypersensitivity. These effects have been described in the introduction and discussion sections.
- Rev #3: The following study support the use of an aluminium lactate/potassium nitrate/hydroxylapatite toothpaste for DH management. please compare your results with the findings of this study.
Seong J, Newcombe RG, Foskett HL, Davies M, West NX. A randomised controlled trial to compare the efficacy of an aluminium lactate/potassium nitrate/hydroxylapatite toothpaste with a control toothpaste for the prevention of dentine hypersensitivity. J Dent. 2021 May;108:103619. doi: 10.1016/j.jdent.2021.103619. Epub 2021 Feb 26. PMID: 33647373.
Answer: We thank the reviewer for the reference. This is a good example about what I just described in the comment above. The pronounced effect of the mixed-HAP particles compared to the high soluble potassium nitrate may be related to the easy of precipitation and adhesion of hydroxyapatite-base particles on the tooth surface. This reference was included in the manuscript.

Reviewer 4 Report
This is a very good paper, well written and reporting a well designed experimental study in which strontium carbonate, a mixed strontium/calcium carbonate and calcium carbonate nanoparticles have been prepared and studied as treatments for dental hypersensitivity. The overall conclusion is that the mixed strontium/calcium carbonate shows promise for this application.
One particularly commendable feature of the manuscript is that the authors have not just assumed that their carbonate compounds are nanoparticulate; they have demonstrated the point experimentally. Their results also show that the mean particle diameters vary with chemical composition, which is an interesting finding.
Despite the overall high quality, there are a few very minor corrections needed before the paper can be accepted. These are as follows:
Line 64: Replace "nervous" with "nerve"
Line 67: The name of the compound is strange, and I think should be "tin monofluorophosphate".
Line 70: Replace "citric" with "citrus".
Line 82: Add the word "the" before "acetate".
Line 88: Move the word "yet" so that it comes between "not" and "been".
Line 111: Replace "Dentistry" with "Dental".
Line 370: Replace "intense" with "intensity".
Line 584: Change "its" to "their".
Line 652: Change the opening sentence to "This study reports the preparation of gels...".
Author Response
Dear Reviewer,
We would like to sincerely thank you for the comments on our manuscript, which were very useful for improving the quality of the text. We have taken all the comments into account in the revised version of the paper. All the changes are highlighted in main the text.
Reviewer 4
This is a very good paper, well written and reporting a well designed experimental study in which strontium carbonate, a mixed strontium/calcium carbonate and calcium carbonate nanoparticles have been prepared and studied as treatments for dental hypersensitivity. The overall conclusion is that the mixed strontium/calcium carbonate shows promise for this application.
One particularly commendable feature of the manuscript is that the authors have not just assumed that their carbonate compounds are nanoparticulate; they have demonstrated the point experimentally. Their results also show that the mean particle diameters vary with chemical composition, which is an interesting finding.
Despite the overall high quality, there are a few very minor corrections needed before the paper can be accepted. These are as follows:
Rev #4: Line 64: Replace "nervous" with "nerve"
Answer: Thank you very much for your comment. The word has been replaced.
Rev #4: The name of the compound is strange, and I think should be "tin monofluorophosphate".
Answer: Thank you very much for point this mistake out. The name of the compound has corrected.
Rev #4: Replace "citric" with "citrus".
Answer: Thank you very much for your comment. The word was replaced.
Rev #4: Add the word "the" before "acetate".
Answer: Thank you very much for your comment. The word “the” was added.
Rev #4: Move the word "yet" so that it comes between "not" and "been".
Answer: Thank you very much for your comment. We changed the position of the word “yet”
Rev #4: Replace "Dentistry" with "Dental".
Answer: Thank you very much for your comment. It was changed.
Rev #4: Replace "intense" with "intensity".
Answer: Thank you very much for your comment. The word “intense” was replaced with “intensity”.
Rev #4: Change "its" to "their".
Answer: Thank you very much for your comment. It was corrected.
Rev #4: Change the opening sentence to "This study reports the preparation of gels...".
Answer: Thank you very much for your comment. However, following the suggestion of another reviewer, in order no make the conclusion section more concise and direct, this brief introductory sentence was deleted from the text.
